# Adapting a peer recovery coach-delivered behavioral activation intervention for problematic substance use in a medically underserved community in Baltimore City

**Emily N. Satinsky**[1]*, **Kelly Doran**[2], **Julia W. Felton**[3], **Mary Kleinman**[1], **Dwayne Dean**[2], **Jessica F. Magidson**[1]

1 Department of Psychology, University of Maryland, College Park, Maryland, United States of America,
2 Department of Nursing, University of Maryland, Baltimore, Maryland, United States of America,
3 Department of Public Health, Michigan State University, Flint, Michigan, United States of America

* esatinsk@umd.edu

## Abstract

Low-income, racial/ethnic minority groups have disproportionately high rates of problematic substance use yet face barriers in accessing evidence-based interventions (EBIs). Peer recovery coaches (PRCs), individuals with lived experience with problematic substance use, may provide an effective approach to reaching these individuals. Traditionally PRCs have focused on bridging to other types of care rather than delivering EBIs themselves. The aim of this study was to assess perceptions of the appropriateness of a PRC-delivered adapted behavioral activation (BA) intervention to reduce problematic substance use for individuals not engaged in care. This study was conducted at a community resource center in Baltimore, Maryland serving low-income and homeless clients who have high rates of problematic substance use yet also face barriers to accessing care. Guided by the ADAPT-ITT framework, we conducted semi-structured key informant interviews with clients ($n = 30$) with past or present problematic substance use, and a focus group with community providers, including staff at the community resource center ($n = 5$) and PRCs ($n = 6$) from the community. Thirty percent ($n = 9$) of clients interviewed reported past problematic substance use and 70% ($n = 21$) met criteria for current use, most commonly cocaine and opioids. Clients, center staff, and PRCs shared that PRC-delivered BA could be acceptable and appropriate with suggested adaptations, including adding peer-delivered case-management and linkage to care alongside BA, and tailoring BA to include activities that are accessible and feasible in the community. These findings will inform the adaptation of PRC-delivered BA to address problematic substance use in this setting.

## Introduction

Maryland ranks in the top five states for most opioid-related deaths [1–3]. The overdose fatality rate in Baltimore specifically is 30% higher than the rest of the state and the highest of any

**Data Availability Statement:** All relevant data are within the manuscript and its Supporting Information files.

**Funding:** This study was fully supported by the University of Maryland Baltimore-University of Maryland College Park Innovation Seed Grant, awarded to JF, KD, and JFM. The funding grant had no role in study design, data collection and analysis, decision to publish, or preparation of the manuscript.

**Competing interests:** The authors have declared that no competing interests exist.

metropolitan county in the United States (US) [4, 5]. Baltimore has a long history of economic and racial inequality [4, 6]; evidence suggests that neighborhood deprivation within Baltimore may increase the odds of substance use disorder (SUD) [7]. Furthermore, recent estimates have indicated widening disparities in drug-related mortality based on socioeconomic status (SES), with lower SES individuals at most risk [8–13]. While National Survey on Drug Use and Health data indicate that 7.9% of Maryland residents met criteria for SUD in 2017 [14], an estimated 44% of people experiencing homelessness in Baltimore City are living with an SUD [15]. Compounding high rates of SUD and associated negative outcomes, low-income, racial/ethnic minority populations, such as those in resource-limited Baltimore neighborhoods, face barriers in accessing and engaging in care [16]. These individuals are often not engaged in the healthcare system and do not seek substance use treatment due to perceived stigma and discrimination, lack of awareness of available services, lack of healthcare coverage, and transportation and other cost barriers [17–20]. Evidence-based interventions (EBIs) that account for unique social and structural stressors are best suited to address the needs of this vulnerable population [21, 22]. However, barriers to access, including restrictions in time, trained personnel, and financial resources of community-based agencies impede the effective dissemination of EBIs to this population [23].

Peer recovery coaches (PRCs), individuals with lived substance use experience and who have gone through the recovery process themselves, may be uniquely suited to address the barriers faced by low-income, racial/ethnic minority individuals with SUD [24–27]. PRCs have scaled in the US recently, and the majority of states now have formal state certification programs [28]. Although PRC roles and training can vary widely across clinical contexts, PRCs have typically been used to support linkage to community-based treatment [29], provide mentorship, support service navigation, promote retention in care, and reduce barriers to engagement [24]. Utilizing PRCs to improve linkage to treatment offers an acceptable, destigmatizing, flexible approach that can address common barriers faced by low-income, racial/ethnic minority individuals, including stigma, structural barriers, housing instability, challenges navigating services, and other psychosocial factors [26]. Furthermore, PRCs from similar demographic backgrounds can improve patient engagement and retention in treatment as a result of relatability and decreased perceptions of stigma [30]. Despite the rapid scale up of PRCs in the US, there remains a lack of clarity regarding their roles within existing clinical and community-based teams, including effective training and supervision models, and scope of practice.

Despite PRCs' established role in recovery support services, there has been little research aimed at evaluating PRC delivery of EBIs specifically. Behavioral activation (BA) is an EBI originally developed as an efficacious treatment for depression [31], which aims to promote engagement in adaptive, valued behaviors and, through this, increase positive reinforcement in one's natural environment. Previous research has demonstrated applicability of BA for substance use and its effectiveness in improving substance use treatment outcomes among low-income, racial/ethnic minority individuals [32–35]. When applied to substance use, BA encourages individuals to identify and schedule activities seen as incompatible with substance use; through these activities, individuals increase activation and positive reinforcement in their environment, develop motivation for change, and disrupt negative behavioral cycles with healthier behaviors. BA is straightforward to deliver and has been shown to be feasible for delivery by lay health workers and nurses in the context of depression [36–40]. As such, BA has specific promise for delivery by PRCs within a recovery support context. Yet, despite empirical support for positive effects of BA on substance use treatment outcomes [41, 42], and evidence to support acceptability for peer or lay counselor-delivery in low- and middle-income

countries [27, 39, 43–51], to date, there has been little research to develop or adapt PRC-delivered BA for problematic substance use in underserved settings in the US.

The first aim of this study was to determine perceptions of the appropriateness of a PRC-delivered adapted BA intervention for a low-income minority population in order to subsequently design and implement an appropriate intervention for problematic substance use. Furthermore, the study aimed to assess barriers and facilitators from multiple perspectives to implementing a PRC-delivered BA intervention in a medically underserved community-based setting for vulnerable and hard-to-reach individuals.

## Methods

This study comprised the first phase of a study aimed at adapting and implementing a PRC-delivered BA intervention for problematic substance use among a low-income, racial/ethnic minority population receiving services in a community-based setting. This formative phase was designed to gather feedback from community stakeholders to inform intervention adaptation. This study used qualitative methods to solicit feedback from the target population, center staff, and PRCs from the community to ultimately adapt BA for PRC delivery in a medically underserved community-based setting. The University Institutional Review Board approved all study procedures.

### ADAPT-ITT

This study was guided by the first three steps of the ADAPT-ITT framework [52]: Assessment, Decision, and Administration. ADAPT-ITT, originally developed for HIV-related EBIs, sequentially lays out steps for intervention adaptation. The first step, "Assessment", involves conducting formative research to determine stakeholders' needs and preferences, and identify implementation barriers and facilitators. The next step, "Decision", involves incorporating stakeholders' feedback into decision-making around which intervention components to include. The third step, "Administration", uses "theater-testing" to pretest intervention components and elicit stakeholder reactions. This framework increases the likelihood that the intervention is acceptable and appropriate for a new setting and target population [52].

### Setting

Qualitative procedures were conducted at a community resource center in West Baltimore serving low-income and homeless clients. The Baltimore-based community resource center is open every weekday from 8:00am to 2:00pm. Based on data routinely collected at the organization, the center estimates that approximately 60% of clients experience homelessness or housing instability. Substance use is also prevalent, with approximately 40% of clients reporting use of alcohol and/or other drugs in the past 24 hours, and approximately 70% of these individuals indicating being "mostly" or "absolutely" ready to seek treatment. Therefore, the resource center provides a unique opportunity to engage vulnerable and hard-to-reach individuals, not otherwise engaged in health care services, in treatment.

### Key informant interview procedures

Recruitment took place from June to September 2018. Research staff discussed the study procedures with center clients, briefly describing aims and explaining the focus on individuals with experiences of substance use. Clients were eligible to participate in semi-structured key informant (KI) interviews if they were over 18, showed no signs of untreated or undertreated major mental illness such as psychosis or mania that would interfere with study procedures,

and demonstrated past or present problematic substance use. For the purposes of this study, we based inclusion criteria on problematic substance use as opposed to a diagnosis of SUD to reflect the priorities in this setting and to increase the reach of the intervention. Problematic substance use was defined based on the "moderate" to "severe" category of the WHO-ASSIST [53], which is a well-validated screening assessment used to identify problematic substance use and determine severity. Moderate to severe problematic use on the WHO-ASSIST reflects use that may present an individual with health, social, or economic problems, which is distinct from but may also reflect a diagnosable SUD. On the WHO-ASSIST, a score of 4 or above for drug use and/or a score of 11 or above for alcohol use indicates moderate to severe problematic substance use. These scores were used as cut points for study eligibility criteria. Individuals were eligible for problematic use of any type of substance (including alcohol, cannabis, cocaine, opioids). For each class of substance on the WHO-ASSIST, participants were asked about lifetime use as well as past-three-month use. The last question on the instrument asks the individual if they have ever injected drugs and if they have injected drugs within the past three months.

If center clients expressed interest in taking part in KI interviews, they were brought to a private space to complete the consent process, including an evaluation of ability to provide informed consent [54]. After the research team member read the consent form aloud, the client was asked to respond to a series of questions meant to guide the evaluation of one's ability to provide consent. Specifically, the client was asked to reiterate (1) what would be expected of him or her, (2) potential risks, and (3) what they would do if they decided they no longer wished to participate [54]. If a client struggled to answer one of these questions, the research team member reviewed the study procedures until the client was able to explain expectations. If the client was unable to adequately answer these questions, even after reviewing the consent form with the research team member, they were determined unable to provide consent. Following the consent procedures, the research team member went through screening questions. If the participant was determined eligible based on the above criteria, the research staff proceeded with the interview. Research staff emphasized the voluntary and confidential nature of participation and informed participants that interviews would be audio-recorded for research purposes only. Two members of the research team were present during each interview; one asked the questions and the other took notes and recorded other observations. Interviews lasted approximately 30 minutes (one hour maximum). Each participant was offered a resource sheet at the end of the interview with information on mental health, physical health, and substance use treatment providers in the area.

Following the Assessment step of the ADAPT-ITT framework [52], interview guides with clients at the community resource center first focused on preferences for working with a PRC, readiness for change, and perceived needs to guide selection of intervention components (Decision step; [52]). The interview questions on readiness for change had participants rate their readiness and motivation for recovery on a scale of 1 to 10, with 10 being "absolutely ready" to take steps in their recovery process. Participants were then asked to elaborate on their rating. When asked for responses to a BA intervention, participants were provided with an adapted Pleasant Events Schedule [55] (see S1 Appendix A). The Pleasant Events Schedule comprises an inventory of potentially reinforcing events and has been widely used in behavioral depression interventions and dialectical behavior therapy [56]. We used an adapted version of the Pleasant Events Schedule that we had previously modified for an underserved, substance using population in a low-income setting [27] and further adapted for this setting. As part of step three of the ADAPT-ITT framework (Administration), participants were asked to identify items from the list, as well as come up with their own activities, that they enjoyed and did not associate with substance use. Participants were also asked how engagement in

those activities could be integrated into their lives and contribute to their recovery. Clients were offered gift certificates as reimbursement for their time ($10). Reimbursement was determined based upon expected time commitment and for consistency with other research conducted at this community site.

## Focus group procedures

In addition to the interviews with center clients, the research team conducted a focus group (FG) with center staff, the PRC hired to implement the pilot trial of this project, and other PRCs working throughout the Baltimore community. The PRC hired for the study helped recruit community PRCs from local agencies, health care organizations, and local peer recovery networks. The focus group took place in August 2018 and lasted approximately two hours. Participants provided written informed consent. The focus group guide focused on assessing barriers and facilitators for a PRC-led modified BA intervention in this setting following the Assessment step of ADAPT-ITT [52] and input on perceived need for intervention components (Decision; [52]). The focus group took place after the first half of client interviews had been conducted; as such, the research team had an initial sense of client responses to the proposed intervention and were able to adapt the focus group questions and structure appropriately. To give staff/PRCs a sense of client participants' initial reactions to BA and working with a PRC, FG participants were provided with a brief summary of these client interview responses. BA treatment components were also described to gauge reactions and input from staff/PRCs, following the Administration step of ADAPT-ITT [52]. FG participants were offered gift certificates as reimbursement for their time ($50). Reimbursement was determined based upon expected time commitment and consistency with other research conducted with staff and PRCs at this and other local community sites.

## Participants

Thirty-three center clients expressed interest in participating in an interview. Of these individuals, one decided she was no longer interested after completing the consent process, and two did not meet eligibility criteria ($n = 1$ did not meet criteria for moderate to severe past or present problematic substance use on the WHO-ASSIST; $n = 1$ showed signs of active psychosis during screening procedures that would interfere with participation in the semi-structured interview).

Client participants ($n = 30$) had a median age of 51 (IQR = 16) and were 70% male, 50% Black, 47% White, and 3% mixed race (Table 1). Most participants held either a high school or equivalent degree (43%, $n = 13$) or did not finish high school (33%, $n = 10$). A third of the client participants reported they were currently working, including part- or full-time employment ($n = 5$), sex work ($n = 2$), or other informal employment ($n = 3$).

Of the client participants, 70% ($n = 21$) reported current problematic substance use and 30% ($n = 9$) met WHO-ASSIST criteria for past problematic substance use (Table 2). Cocaine and opioids were the most problematic substances (median WHO-ASSIST scores of 29.5 (IQR = 18.4) and 33.5 (IQR = 7.5) respectively). Across all 30 participants, 77% ($n = 23$) reported lifetime problematic cocaine use and 70% ($n = 21$) reported lifetime problematic opioid use. Thirty-eight percent ($n = 8$) of participants with current problematic substance use had injected drugs in the past three months. Among client participants reporting current problematic substance use, poly-substance use was common: 52% ($n = 11$) met criteria for moderate to severe problematic substance use for three or more substances and 19% ($n = 4$) met criteria for moderate to severe problematic substance use for two substances. Forty-seven percent ($n = 14$) of the client participants were on or had previously been prescribed

**Table 1. Demographic characteristics of qualitative participants.**

| | Key Informant Participants—Center Clients (*n* = 30) | Focus Group Participants | |
| --- | --- | --- | --- |
| | | Community PRCs (*n* = 6) | Center Staff (*n* = 5) |
| **Median Age** (years) | 51 (IQR: 16) | 49.5 (IQR: 15) | 34 (IQR: 25.5) |
| **Gender** (male) | 70% (*n* = 21) | 50% (*n* = 3) | 40% (*n* = 2) |
| **Race** | 50% (*n* = 15) | 83% (*n* = 5) | 20% (*n* = 1) |
| Black or African American | | | |
| White | 47% (*n* = 14) | 17% (*n* = 1) | 60% (*n* = 3) |
| Asian | NA | NA | 20% (*n* = 1) |
| Mixed Race | 3% (*n* = 1) | NA | NA |
| **Education** | | | |
| Less than high school | 33% (*n* = 10) | 17% (*n* = 1) | NA |
| High school or equivalent | 43% (*n* = 13) | 33% (*n* = 2) | NA |
| Some college, no degree | 20% (*n* = 6) | 50% (*n* = 3) | NA |
| Bachelor's degree | 3% (*n* = 1) | NA | 60% (*n* = 3) |
| Graduate degree | NA | NA | 40% (*n* = 2) |

medication for opioid use disorder (MOUD), including methadone (*n* = 14) and buprenorphine/Suboxone (*n* = 5).

When client participants endorsing current problematic substance use were asked about their readiness for change, 76% (*n* = 16) of participants provided a numerical response on the readiness ruler. The median readiness rating among responders was a 9.25/10 (IQR: 6.5). The remaining participants with current problematic substance use struggled to quantify their readiness for change and elected not to provide a numerical rating (*n* = 5).

FG participants (*n* = 11) included center staff (*n* = 5) and community PRCs (*n* = 6). Staff/PRC participants had a median age of 47 (IQR = 25) and were 45% male, 55% Black, 36% White, and 9% Asian (Table 1). Staff members included site leadership (*n* = 2), a program director (*n* = 1), and case managers (*n* = 2). Community PRCs had experience in a range of substance use treatment environments including the emergency department of a local hospital (*n* = 3), a prison (*n* = 1), a community resource center (*n* = 1), and a mental health and substance use treatment center (*n* = 1).

## Data analysis

All recordings were transcribed, and all transcriptions were double-checked for accuracy. Using thematic analysis, separate KI interview and FG codebooks were iteratively developed to outline themes, sub-themes, and definitions in the transcripts. These codebooks were modified as new concepts arose [57]. Using these established codebooks, two independent coders coded

**Table 2. WHO-ASSIST substance use risk category among key informant interview participants.**

| | Participants with Current Problematic Substance Use (*n* = 21) | | | Participants with Past Problematic Substance Use (*n* = 9) | | |
| --- | --- | --- | --- | --- | --- | --- |
| | No/Low Risk | Moderate Risk | Severe Risk | No/Low Risk | Moderate Risk | Severe Risk |
| **Alcohol** | 57.1% (*n* = 12) | 14.3% (*n* = 3) | 28.6% (*n* = 6) | 11.1% (*n* = 1) | 55.6% (*n* = 5) | 33.3% (*n* = 3) |
| **Cannabis** | 57.1% (*n* = 12) | 38.1% (*n* = 8) | 4.8% (*n* = 1) | NA | 77.8% (*n* = 7) | 22.2% *n* = 2) |
| **Cocaine** | 28.6% (*n* = 6) | 23.8% (*n* = 5) | 47.6% (*n* = 10) | 11.1% *n* = 1 | 55.6% (*n* = 5) | 33.3% (*n* = 3) |
| **Opioids** | 23.8% (*n* = 5) | 14.3% (*n* = 3) | 61.9% (*n* = 13) | 44.4% (*n* = 4) | 22.2% (*n* = 2) | 33.3% *n* = 3) |

*Based on established WHO-ASSIST categorization [53]

transcripts using Nvivo Version 11. The coders met weekly to discuss and resolve discrepancies in coding. A third person arbiter was involved in these meetings. The coders achieved high inter-coder reliability ($\kappa > 0.90$).

## Results

Six intersecting themes emerged from the KI interviews and FG following our two study aims: 1) to assess perceptions of the appropriateness of the proposed treatment approach; and 2) to identify barriers and facilitators to feasible and effective implementation. Participants expressed (1) a preference for working with a PRC; and (2) perceived appropriateness of BA with some adaptations as an approach that may support individuals with problematic substance use in this setting (Aim 1). Participants pointed to barriers to implementation of the proposed approach, including: (3) structural and environmental factors (such as housing instability, high crime neighborhoods) and (4) the severity of clients' substance use (Aim 2 – Barriers). Participants provided input on how to navigate these barriers, namely by (5) incorporating PRC-delivered case management alongside BA and (6) integrating PRC-supported linkage to MOUD (Aim 2 –Facilitators).

### Perceived appropriateness of a PRC-delivered BA intervention

**Preference for working with a PRC.** Overall, participants described their perceived appropriateness of a PRC-delivered intervention in this setting. Specifically, several client participants had previously worked with a PRC and described how working with someone who is "experience-taught versus book-taught" [Client KI Participant, (C1)] increased their level of comfort in sharing experiences and obstacles. Staff/PRC participants similarly noted the importance of PRCs in recovery, with PRCs invoking their experiences working with peers during their own recovery processes and working as peers themselves. Two community PRCs explained:

"Because they've got a story, and the person can identify."

- - [PRC FG Participant, (P1)]

"I'll start to disclose or share part of my story then. And it kinda, it pretty much draws them in. Cause now, it's, there's no hierarchy. We're on a level plane."

- - [PRC FG Participant, (P2)]

Among those who had not previously worked with a PRC, client participants stated the importance of working with someone who had lived through similar experiences. One participant remarked, "I think they would be a very strong role in my recovery" [Client KI Participant, (C2)].

Stressing the importance of peer support in recovery, some client participants cited experiences in substance use treatment programs where they felt judged by counselors who did not have experiences with problematic substance use or homelessness. C1 explained:

"You know like, I've had a couple. . .counselors, who've never [used drugs]. I've always felt like they're looking down their noses at me."

This participant noted that if he had worked with a PRC who had been in similar situations, he believed he could have found common ground on which to relate and open up.

In addition to the role PRCs can play in building rapport over commonalities, client participants also discussed PRCs' role in increasing motivation to change. Several clients explained that a major facilitator in the recovery process is seeing it work for someone else. Client participants repeatedly stressed the value of hearing someone else relate their path to recovery in supporting the clients' own readiness for treatment. "It helps you," one participant remarked, "to see that it can actually be done" [Client KI Participant (C3)].

**Perceived appropriateness of BA in this setting.** Clients stressed how rewarding, substance-free activities may keep them from using or help them cope with urges to use alcohol and/or other drugs. Specifically, participants explained that BA could work by keeping them busy with positively reinforcing activities. Several participants mentioned that when they are busy with jobs or enjoyable substance-free activities, they experience fewer urges to use. Two participants explained:

"I have no time for alcohol and drugs . . .if I did like um, five of these things, and work—you know what I mean? Wouldn't have time to do. . .drugs."

- - [Client KI Participant (C4)]

"Just working, period. . .. When I keep myself busy . . .I don't think about using . . .That's my biggest issue, is staying busy. Like, so when I do have a job, I try to work as late as possible. Keep myself busy, because then I'm not worried about using substances. When I have all my spare time and there's nothing to do, that's when I'm like . . . let's go get high."

- - [Client KI Participant (C5)]

In addition to the importance of physically staying busy, one participant explained that "[applying] my mind to something else" helps him cope with urges [Client KI Participant (C6)].

Client participants mentioned a range of activities that they consider distinct from and incompatible with substance use. Of the different activities mentioned, sports and exercise (such as walking, working out, soccer, dancing) and religious activities (such as reading the Bible, listening to Gospel music, going to church, praying) were discussed most frequently. Discussing exercise, one participant explained:

"The exercising, that actually gives you the dopamine, uh, rush as well. So that, other than mentally, you physically, you got something physically. That makes you feel good too."

- - [Client KI Participant (C7)]

In line with C7's response, other participants described how spending time outside and being active helps them stay occupied and avoid cravings.

Participants similarly discussed engagement in religious activity as being incompatible with substance use. Participants described:

"Because I'm very spiritual, and I just don't, I would never like smoke and read the Bible. Like, that's just, like too different."

–[Client KI Participant (C8)]

"When you reading the Bible. . .it don't have to be just alcohol. . .any situation that you got, once you start reading that Bible. . .it's a cure for everything you know."

- - [Client KI Participant (C9)]

### Barriers to a PRC-delivered BA intervention

**Structural and environmental factors.** Participants provided feedback on barriers to effective delivery of a PRC-led BA intervention. Client and staff/PRC participants described a lack of stability in clients' lives as a major barrier to successfully implementing PRC-delivered BA. In particular, staff/PRC participants identified pervasive structural barriers, for example homelessness and housing instability, as limiting clients' ability to successfully engage in BA. One center staff member summarized this barrier:

> "And the um, the lack of that stability tends to be a huge hindrance to pursuing treatment. I think a lot of people tend to have like their world's problems . . . Are too big, or are in the way for them to actually start like go away from them to then work with, you know, recovery."
>
> –[Center Staff FG Participant (S1)]

As this participant explained, clients may not have the flexibility to focus on incorporating positive reinforcing activities into their daily life when they are consumed by the "world's problems."

Alongside, and as a result of, the structural barriers associated with housing and financial instability, clients discussed feeling unsafe in their environment, which is a barrier to engagement and successful outcomes in any type of intervention. When there's a "needle here and a needle there," as one participant described the constant reminder of drugs and drug paraphernalia [Client KI Participant (C10)], sustaining motivation for recovery and engaging in a BA intervention becomes significantly more difficult. Another participant reiterated this point:

> "I'm gonna be truthful . . .because see, like everywhere it's people getting high. . .drugs, there's drugs, and drugs will find you if you don't find it, and that's how it goes."
>
> –[Client KI Participant (C11)]

In addition to living in an environment rife with substance use and exposure to drugs, participants repeatedly mentioned pervasive violence and crime in the neighborhood, which needs to be considered when adapting a BA intervention. C5 explained,

> "I don't feel safe when I'm outside anymore. People getting shot all over the place. . . You got to constantly watchin' over your back. . .at least once a week. . .people trying to rob me."

As client participants repeatedly described, the prevalence of drug use and drug cues as well as high levels of crime may impede their ability to incorporate certain activities into their daily lives.

**Severity of clients' substance use.** The severity of substance use within this population repeatedly arose as a barrier to clients feeling ready to engage in BA or incorporate positive reinforcing activities into their daily lives. Multiple client participants stressed that when they are in the throes of addiction, it becomes nearly impossible to find the motivation to engage in certain activities. After being asked about activities he enjoys doing that don't involve drugs or alcohol, C1 noted:

> "When I'm getting high, none of that even comes into play. When you're living this life, it's the only thing that there is. You know. That's the problem with, with drugs, they completely

take over your everything. You know, all the stuff you used to like to do, doesn't even exist. It's all a big blur of the past."

Another participant similarly described: "Because, [the] mind is so, so set on, set on getting high. . ... They ain't got time to read a book." (C9)

Particularly among individuals using opioids, participants described beliefs that a BA intervention alone would not be sufficient (for example without medication and/or other support to reduce or stop using), and that engaging in positive reinforcing activities would be a challenge in the context of heavy use and the cycle of withdrawal. For instance, one participant noted that when people are experiencing withdrawal, "all they want to do is lay down in bed" [Client KI Participant (C12)].

As these client participants and other staff/PRC participants expressed, the severity of problematic substance use within this population presents serious barriers to engaging clients in a brief, structured PRC-delivered BA intervention.

### Facilitators for a PRC-delivered BA intervention

In light of concerns around integrating a PRC-delivered BA intervention for problematic substance use into this setting, client and staff/PRC participants suggested solutions for how to address the noted barriers. Particularly, clients and staff/PRCs noted that incorporating PRC-delivered case management and wraparound services, as well as referral and linkage to MOUD, alongside BA would improve the appropriateness, feasibility, and potentially the effectiveness of a PRC-delivered BA intervention.

**PRC-delivered resource- and case-management.**   Several staff/PRCs noted that in the context of pervasive instability among clients, a PRC-delivered BA intervention may only be effective alongside additional support for resource and case management. Staff/PRCs presented the importance of addressing basic needs, including acquisition of identification, shelter, food, and clothing, through PRC-delivered resource- and case-management in order to promote engagement in BA. Staff/PRC participants described the value of acting as a bridge to other services:

"Keep them linked up. . .to their treatment and whatever other type of assistance I can help them with. Like ID, maybe a job lead, maybe housing, food, etc."

–[PRC FG Participant (P3)]

Staff/PRCs shared that once those basic needs are met, it may allow for greater engagement in a BA intervention. As shared by one staff participant, "Fulfill those needs, there'd be a lot more room. . .to build from." (S1)

Clients similarly noted that they thought an intervention would be most effective if it followed or took place in conjunction with linkage to other resources and support systems. One client, for instance, shared that "a good PRC would be able to direct them into [housing] resources" (C7).

**PRC-supported linkage to MOUD.**   PRC-supported linkage to MOUD was one such resource that participants described could facilitate a more effective BA intervention for individuals using opioids. Given the severity of clients' problematic substance use, clients commented that a BA intervention should be delivered alongside MOUD. One participant, for example, explained that a PRC-delivered BA intervention would be best if it involved, "the medicine and then the talk" (C3). Another client reiterated this claim:

"You'd have to have medication in it. Cause the simple fact, you got people that get high, you know, and they ain't trying to hear nothing until they get, if you, if you can get a person to relax, and open up, you got the first step going."

(C6)

As these participants explained, there would be more opportunity to deliver a BA intervention and successfully engage participants in substance-free positive reinforcement if a PRC could first link a client to MOUD and support them to get stabilized before initiating a behavioral intervention.

## Discussion

Findings from this study highlight the perceived appropriateness of a PRC-delivered approach and BA specifically to address problematic substance use in this community setting. Despite positive responses, clients and staff/PRCs presented barriers to consider when adapting a PRC-delivered intervention. Specifically, participants explained that the lack of structural and psychosocial stability in the lives of center clients might impede feasibility and effective implementation of a brief behavioral intervention. In response to these barriers, participants suggested implementing BA alongside PRC-delivered case management and linkage to MOUD.

Overall, participants described a preference for a PRC-delivered intervention, demonstrating the likelihood that it will be acceptable to incorporate peer work into the community setting. Positive responses to working with a peer are in-line with the expanding role of peers to address problematic substance use in the US [58]. A study conducted in Baltimore demonstrated the feasibility, efficacy, and sustainability of using peers to support HIV-prevention and harm reduction efforts among people who inject drugs [59]. Peers were able to build rapport based on commonalities and promote harm reduction efforts in the community. Studies conducted in emergency departments and clinical settings in the US and in other contexts globally have similarly reflected the acceptability and potential for peers in harm reduction and recovery support services, including for reduction of use, decreased risk behaviors, and improved quality of life [26, 60–67]. Particularly considering some stigmatizing interactions with health care workers, as one client noted above, individuals may be better able to share experiences and address underlying factors contributing to problematic substance use when working with a PRC. An important consideration when screening and hiring a PRC is the individual's personal history and path to recovery; specifically, some PRCs may not support MOUD or other medication-based treatment models [26]. As such, identifying PRCs that are open to multiple pathways to recovery is essential.

In addition to the stated preference for working with a PRC, participants were able to identify positive, substance-free activities necessary for successful BA implementation. Some client participants described the role of "staying busy" in curbing cravings and preventing relapse. While there is some research to suggest that daily routinization and scheduling (or staying busy) can be beneficial for substance use outcomes, the conceptual model for BA encourages identification of activities that are uniquely rewarding and values-driven for the individual [68, 69]. Client participants listed a range of positive reinforcing activities that they felt are distinct from substance use. While participants listed several activities, such as spending time with friends and family, volunteering, and arts and music, two activity categories arose as most salient: sports/exercise and religious activities. These specific categories are consistent with research describing the role of religious activity, including associated spirituality and social support [70–73], and sports and exercise [74, 75] in supporting recovery from problematic substance use. Furthermore, these reflect the reinforcing, substance-free activities listed in

qualitative interviews among a population with problematic substance use in a South African HIV care setting [44], indicating the potential role of these activities in BA for problematic substance use across cultures. A recent systematic review demonstrated an inverse relationship between substance-free positive activities and substance use [76], further demonstrating potential for BA in this setting, particularly if tailored to participants' preferences and activity selection that is feasible and accessible in this community.

Despite acknowledging substance-free, rewarding activities that may be useful in disrupting the behavioral cycle of addiction, participants described important barriers that may interfere with engaging in or initiating some of these activities. For instance, participants described pervasive structural and psychosocial instability. As the community resource center primarily serves clients facing homelessness or housing instability, several clients, staff, and PRCs reflected that it would be difficult for clients to engage in a brief BA intervention and incorporate positive activities into their lives while struggling with financial stressors. Similarly, several client participants discussed feeling unsafe in their current environment. This feeling may further impede clients' ability to incorporate certain activities into their day-to-day lives (for example feeling unsafe playing sports outside due to high crime neighborhoods). Socioeconomic barriers as well as living in violent, high crime communities with open-air drug markets are linked with high rates of problematic substance use [77, 78] and warrant adaptation of behavioral interventions to address these barriers. When adapting a BA intervention specifically, it is important to acknowledge community-level factors and resource constraints in identifying and scheduling activities, with a focus on unique risk factors for relapse, safety when engaging in activities, and identifying individualized activity scheduling plans that take these risk factors into account [32, 35, 79, 80].

The severity of clients' problematic substance use repeatedly arose as another barrier to the effective implementation of a PRC-delivered BA intervention. Although we did not limit recruitment to clients with opioid use disorder (OUD), the majority of clients interviewed met criteria for past or present problematic opioid use. As MOUD is the gold standard, efficacious approach for treating OUD [81, 82], it is understood that MOUD is an important component of treatment for these individuals. Without medication to treat OUD and curb cravings and withdrawal symptoms, clients with OUD may have difficulties changing behavioral cycles and increasing substance-free, rewarding activities. A clear recommendation was the need to integrate BA with MOUD.

To address barriers to engaging this population in care, particularly structural and psychosocial instability, client and staff/PRC participants suggested combining PRC-delivered BA with PRC-delivered case management and linkage to care support, including linkage to services that focus on wraparound care such as attainment of housing and other benefits. As a significant portion of this population is not otherwise engaged in health care services, many individuals may need additional support. Furthermore, considering the high rate of homelessness and housing instability seen among this population, individuals may face additional barriers to engaging in care and incorporating positive reinforcing, values-driven activities into their daily lives. By task sharing case management services with PRCs and incorporating linkage to care alongside PRC-delivered BA, an adapted intervention could have the unique potential to simultaneously increase engagement in healthcare and address probelmatic substance use among hard-to-reach individuals. Among individuals concurrently experiencing homelessness and severe problematic substance use, linkage may involve connection with intensive outpatient treatment programs that include a residential housing component and support attainment of permanent housing following treatment completion. Subsequent phases of this work will test whether peers can indeed take over case management reponsibilities while maintaining ongoing support from and linkage channels with specialized case management services

for more severe concerns. Finally, importantly for patients with OUD, PRC-delivered linkage to care could include referring clients to MOUD, as well as other formalized psychosocial treatment programs for co-occurring substance use and mental health problems.

Given the extensive barriers faced by this population and the high likelihood for relapse, it is important to offer low-threshold, harm reduction approaches to minimize barriers to engaging this population in services to improve their health and well-being. If incorporated with case management and linkage to care, the PRC-delivered BA approach may be uniquely suited to improve quality of life and reduce harms associated with problematic substance use. Rather than solely encouraging patients to stop using substances, the proposed intervention model would emphasize connection with resources and external treatment modalities, engagement with destigmatizing peer providers, and increasing alternative, valued activities that are incompatible with substance use.

These qualitative results are informing an intervention package (see S1 Fig), the effectiveness and implementation of which is currently being evaluated. Given prior work pointing to the effects of BA on improving substance use treatment retention [35], an important next step is to understand how to incorporate PRC-delivered BA with MOUD and other psychosocial treatment modalities to support initiation and retention in care. As retention in MOUD and other higher level services is important for preventing relapse [83], integrating concurrent PRC-delivered BA may facilitate better treatment outcomes and is an important next step of this program of research. In the ongoing research guided by these qualitative results, we aim to explore how PRCs can support delivery of psychosocial treatment, under the close supervision and monitoring of licensed mental health professionals, to support improved MOUD outcomes.

## Limitations

These results need to be considered in the context of existing limitations. KI interviews and the FG were limited by small sample sizes. However, as recommended in qualitative analysis [84], we stopped collecting qualitative data once we reached theoretical saturation—client participants reflected a range of ages, races, levels of substance use severity, and perspectives; center staff were recruited across a spectrum of roles; and community PRCs were recruited from diverse community care settings. While some staff/PRC participants may have felt uncomfortable stating their thoughts in an open forum, all participants spoke and provided feedback during the FG.

As client participants were not going through a BA intervention at the time of the interviews, responses do not reflect experience with the actual intervention model. Rather, we were only able to expose participants to basic elements of BA to elicit feedback. It will be important to assess the acceptability and feasibility of this intervention approach as participants engage in the PRC-delivered BA intervention.

Lastly, it was often hard to tease apart barriers to substance use treatment in general and barriers to BA specifically. Participants listed several challenges to incorporating substance use treatment into this community-based, high-risk setting; however, they did not always make direct links to barriers associated with engagement in positive reinforcing activities. Yet, many of the structural and environmental barriers described by participants would impede participation in certain activities.

## Conclusions

It is an urgent national priority to improve access to substance use treatment and identify strategies that are sustainable and cost-effective to engage and retain hard-to-reach individuals in

care [85]. Qualitative feedback suggests that, with appropriate adaptation, a PRC-delivered BA intervention may be effective in this mission. A PRC-delivered BA intervention delivered alongside peer-facilitated case management and linkage to care services, including linkage to MOUD, has the potential for significant public health impact.

## Supporting information

**S1 Fig. PRC-delivered EBI based upon formative qualitative work.** S1 Fig depicts the three intersecting components of the proposed PRC-delivered intervention treatment approach, based upon formative feedback from the KI interviews and FG. To feasibly and effectively engage and retain a hard-to-reach population, and concurrently address structural and psychosocial barriers to treatment, the adapted PRC-delivered intervention developed based on the formative work presented here will incorporate: behavioral activation to increase substance-free positive reinforcement; linkage to MOUD to reduce cravings and prevent future relapse; and case management to address structural and environmental barriers.
(PDF)

**S1 Appendix. Pleasant Events Schedule adapted for setting.**
(DOCX)

## Acknowledgments

We thank the community resource center and study participants, as well as Morgan Anvari, Samantha Cohen, Kayla Griffin, Alexandra Rose, Hannah Tralka, and Christine Wan for their assistance during interview transcription and coding.

## Author Contributions

**Conceptualization:** Emily N. Satinsky, Kelly Doran, Julia W. Felton, Mary Kleinman, Jessica F. Magidson.

**Data curation:** Emily N. Satinsky, Mary Kleinman.

**Formal analysis:** Emily N. Satinsky, Mary Kleinman, Jessica F. Magidson.

**Funding acquisition:** Julia W. Felton, Jessica F. Magidson.

**Investigation:** Emily N. Satinsky, Kelly Doran, Julia W. Felton, Dwayne Dean, Jessica F. Magidson.

**Methodology:** Emily N. Satinsky, Kelly Doran, Julia W. Felton, Mary Kleinman, Dwayne Dean, Jessica F. Magidson.

**Project administration:** Emily N. Satinsky, Dwayne Dean, Jessica F. Magidson.

**Supervision:** Kelly Doran, Julia W. Felton, Mary Kleinman, Jessica F. Magidson.

**Validation:** Kelly Doran.

**Writing – original draft:** Emily N. Satinsky.

**Writing – review & editing:** Kelly Doran, Julia W. Felton, Mary Kleinman, Dwayne Dean, Jessica F. Magidson.

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
