## [Decision Letter · Decision Letter 0]

1 Oct 2019

PONE-D-19-16724

Adapting a community-based peer recovery coach-delivered behavioral activation intervention for problematic substance use in a medically underserved community in Baltimore City

PLOS ONE

Dear Dr Magidson,

Thank you for submitting your manuscript to PLOS ONE. After careful consideration, we feel that it has merit but does not fully meet PLOS ONE’s publication criteria as it currently stands. Therefore, we invite you to submit a revised version of the manuscript that addresses the points raised during the review process.

We would appreciate receiving your revised manuscript by Nov 15 2019 11:59PM. To enhance the reproducibility of your results, we recommend that if applicable you deposit your laboratory protocols in protocols.io, where a protocol can be assigned its own identifier (DOI) such that it can be cited independently in the future. For instructions see: http://journals.plos.org/plosone/s/submission-guidelines#loc-laboratory-protocols

We look forward to receiving your revised manuscript.

Kind regards,

Bronwyn Myers

Academic Editor

PLOS ONE

Journal Requirements:

'The funders had no role in study design, data collection and analysis, decision to publish, or preparation of the manuscript.'

Please provide an amended Funding Statement that declares *all* the funding or sources of support received during this specific study (whether external or internal to your organization) as detailed online in our guide for authors at http://journals.plos.org/plosone/s/submit-now.  

Please state what role the funders took in the study.  If any authors received a salary from any of your funders, please state which authors and which funder. If the funders had no role, please state: "The funders had no role in study design, data collection and analysis, decision to publish, or preparation of the manuscript."

Additional Editor Comments (if provided):

This is a very interesting submission that addresses an important topic of relevance to improving access to substance abuse interventions. We have now received feedback from two reviewers. While generally enthusiastic, they both identify aspects of the manuscript that require clarification or revision.

I hope you find their detailed comments and recommendations for revision useful and I look forward to receiving a revised version of this manuscript.

Reviewers' comments:

Reviewer's Responses to Questions

**Comments to the Author**

1. Is the manuscript technically sound, and do the data support the conclusions?

Reviewer #1: Yes

Reviewer #2: Yes

2. Has the statistical analysis been performed appropriately and rigorously? 

Reviewer #1: I Don't Know

Reviewer #2: N/A

3. Have the authors made all data underlying the findings in their manuscript fully available?

Reviewer #1: No

Reviewer #2: Yes

4. Is the manuscript presented in an intelligible fashion and written in standard English?

Reviewer #1: Yes

Reviewer #2: Yes

5. Review Comments to the Author

Reviewer #1: This study examines a critical issue of reducing drug use and potentially promoting drug treatment, especially medication for opioid use disorder in a high need and highly impoverished population.

What was the focus of the types of substance use? Although the authors report that “cocaine and opioids were the most problematic,” was it both alcohol and illicit drugs? Was the focus people who use drugs, opioids, or injection drug use? In different sections of the manuscript, there appears to be a different focus.

The authors should provide a more nuanced perspective on recovery coaches. Some have strong ideas on effective approaches, and some do not support medication-assisted treatment. Although it is highly probably that lay health workers are effective for addressing substance use, the references provided for LMIC focus on mental health and HIV rather than substance use.

The authors should describe in detail and provide in the appendices their pleasant events scheduled since the original one cited in the manuscript is very inappropriate for this population.

Provide information on the amount of reimbursement also provide more details on the focus group participants for the peer recovery coaches.

Given the huge differences in environments between impoverished and more affluent communities, if the focus of the study was Behavioral Activation, what were the recommendations and findings on how such an approach can be adapted to urban inner-cities with high levels of crime, open-air drug markets, drug use within families, low levels of employment, and few economic resources. Some of this material appears in the quotes, but it is a critical factor for adapting Behavioral Activation to this population using peer recovery coaches.

One of the themes was “staying busy” but this seem different, albeit some overlap, with pleasant events.

Was “case management” a theme mentioned by clients as well as staff? The quote by a client suggests that they needed resources and support. Why would this need to be case management? How much of this could be done by peer recovery coaches without case managers? Isn’t one of the rationales for peer recovery coaches to reduce costs?

Reviewer #2: This is an interesting piece of qualitative research and is well-written. Addressing the following concerns would strengthen the article.

Introduction:

-it would be valuable at some point in the introduction to provide a definition of problematic substance use and substance use disorders, and any prevalence data (in addition to numbers of opioid-related deaths)

-Line 55: state some of the barriers that literature has shown that people who use substances face in accessing care.

-Behavioral activation: it will be useful to provide some more information about the process and goals of behavioural activation so that it clearly links up to some of the activities in the methods section such as the completion of the readiness to change and PES exercises.

Methods:

-this article is an adaption of existing evidence-based interventions. Did the authors consider the use of a framework to guide this process?

-since the ASSIST measures substance use in the previous three months, expand on how past and present problematic substance use were defined in this study.

-evaluation of ability to provide consent is important, could this be expanded upon in a sentence (what did this process look like?)

-There a few acronyms that are only used once or twice in the article such as DBT, LTC and OUD. It would be easier to read if these were spelled out in the manuscript.

-Explain why the focus group took part after half of the interviews had been completed

-What were the value of the gift certificates provided for reimbursement.

-Please clarify when participants were screened for eligibility and why one did not meet eligibility criteria AFTER consenting took place.

-Think about possibly providing a table to summarise the KI and FG participant characteristics. Also, since opioid use is clearly an issue in this setting, can the authors provide the proportion of problem opioid use in general (in addition to the average score on the ASSIST?)

Results:

-Can the authors indicate when the findings and quotes were from the FGs and when they were from the KI interviews in the results section?

-Some of the quotes back up the same argument and can be put together without further text in between. For example, line 254 on page 12 and lines 282-283 on page 13 can be removed.

-line 265-267 on page 12: rather use such as than abbreviations i.e. and etc.

-The voice of the PCRs seem to be somewhat missing in the quotes section, are there any quotes are any interesting points that they raised specifically as opposed to other staff in the centers?

-P17 line 364-I am not sure that this quote is necessary.

P17 line 368-not entirely sure what this quote is saying-are there any other quotes where the types of resources are discussed that the authors could substitute here?

Discussion:

-P18 line 399: "Yet, clients and staff/PRCs presented barriers to consider." This sentence feels incomplete.

-P19: Are there other studies conducted with PRCs outside of Baltimore than can be references here?

-P19 line 420: There are a number of studies that look at physical activity and religion as prosocial activities that are alternatives to substance use including studies conducted in the US and globally. Could the authors do some additional research and reference some of these?

-It is interesting that the majority of the participants are homeless-how will this affect their access to engagement in interventions, as this is different to living in economically disadvantaged areas with high levels of crime.

-Since there are very likely participants that had levels of substance use that placed them at high levels of risk, it is possible that they may need more formalised psychosocial treatment in addition to pharmacological assistance. This is something for the authors to consider in the discussion.

6. PLOS authors have the option to publish the peer review history of their article (what does this mean?). If published, this will include your full peer review and any attached files.

Reviewer #1: No

Reviewer #2: Yes: Dr. Tara Carney

---

## [Author Response · Author response to Decision Letter 0]

16 Nov 2019

Editor Comments:

This is a very interesting submission that addresses an important topic of relevance to improving access to substance abuse interventions. We have now received feedback from two reviewers. While generally enthusiastic, they both identify aspects of the manuscript that require clarification or revision. I hope you find their detailed comments and recommendations for revision useful and I look forward to receiving a revised version of this manuscript.

RESPONSE: We appreciate this positive feedback on our manuscript. We have incorporated both reviewers’ suggestions for revision and clarification, and we believe these changes have significantly improved our manuscript. We are grateful for the feedback and for this opportunity to resubmit our revised manuscript.

Reviewer #1: 

1. This study examines a critical issue of reducing drug use and potentially promoting drug treatment, especially medication for opioid use disorder in a high need and highly impoverished population. What was the focus of the types of substance use? Although the authors report that “cocaine and opioids were the most problematic,” was it both alcohol and illicit drugs? Was the focus people who use drugs, opioids, or injection drug use? In different sections of the manuscript, there appears to be a different focus.

RESPONSE: Thank you for the opportunity to clarify the substances used in this population and our inclusion criteria. We did not limit our focus to a specific class of substance. Despite the high prevalence of opioid use in Baltimore, we left our eligibility criteria open to problematic use of any type of substance (including alcohol) to incorporate perspectives of all individuals using substances at this site. This decision also reflects the priorities of our community partner to meet the needs of individuals with problematic substance use regardless of type of use. 

As described under “Participants” in the Methods section (p. 11), participants reported most severe problematic use of cocaine and opioids; however, polysubstance use was also common, with many participants also reporting at least moderate severity use of alcohol and cannabis. We provide a breakdown of severity level for commonly reported substances in Table 2 on p. 12. We now also elaborate in the Methods on p. 7-8 under “Key informant interview procedures”: 

“Individuals were eligible for problematic use of any type of substance (including alcohol, cannabis, cocaine, opioids). For each class of substance on the WHO-ASSIST, participants were asked about lifetime use as well as past-three-month use. The last question on the instrument asks the individual if they have ever injected drugs and if they have injected drugs within the past three months.” 

2. The authors should provide a more nuanced perspective on recovery coaches. Some have strong ideas on effective approaches, and some do not support medication-assisted treatment. 

RESPONSE: Thank you for this important point. We agree and have found that the individual peer recovery coach’s attitudes on effective approaches is an important consideration when hiring and training peers. We have provided more detail about this important point in the Discussion (see below). We also agree that more nuanced perspectives on the role of recovery coaches, including the lack of clarity on their training and supervision, is important to highlight, and we have now also provided more detail on this point. 

p. 4: “PRCs have scaled in the US recently, and most states now have formal state certification programs. Although PRC roles and training can vary widely across clinical contexts, PRCs have typically been used to support linkage to community-based treatment, provide mentorship, support service navigation, promote retention in care, and reduce barriers to engagement.” 

p. 23: “An important consideration when screening and hiring a PRC is the individual’s personal history and path to recovery; specifically, some PRCs may not support MOUD or other medication based treatment models. As such, identifying PRCs that are open to multiple pathways to recovery is essential.”

3. Although it is highly probable that lay health workers are effective for addressing substance use, the references provided for LMIC focus on mental health and HIV rather than substance use.

RESPONSE: We have updated the references here (p. 5) to reflect studies focused on the role of peers and lay health workers in substance use treatment in LMICs. Indeed, most research in LMICs on lay health worker-delivered behavioral health models focus on HIV and mental health rather than substance use. However, we have now incorporated existing research that focuses on lay health worker delivered models for alcohol and other substance use. 

Patel V, Weobong B, Nadkarni A, Weiss HA, Anand A, Naik S, et al. The effectiveness and cost-effectiveness of lay counsellor-delivered psychological treatments for harmful and dependent drinking and moderate to severe depression in primary care in India: PREMIUM study protocol for randomized controlled trials. Trials. 2014;15(101).

Papas, R. K., Sidle, J. E., Gakinya, B. N., Baliddawa, J. B., Martino, S., Mwaniki, M. M., ... & Ojwang, C. (2011). Treatment outcomes of a stage 1 cognitive–behavioral trial to reduce alcohol use among human immunodeficiency virus‐infected out‐patients in western Kenya. Addiction, 106(12), 2156-2166.

Papas, R. K., Sidle, J. E., Martino, S., Baliddawa, J. B., Songole, R., Omolo, O. E., ... & Owino-Ong’or, W. D. (2010). Systematic cultural adaptation of cognitive-behavioral therapy to reduce alcohol use among HIV-infected outpatients in western Kenya. AIDS and Behavior, 14(3), 669-678.

Parry, C. D., Morojele, N. K., Myers, B. J., Kekwaletswe, C. T., Manda, S. O., Sorsdahl, K., ... & Shuper, P. A. (2014). Efficacy of an alcohol-focused intervention for improving adherence to antiretroviral therapy (ART) and HIV treatment outcomes–a randomised controlled trial protocol. BMC infectious diseases, 14(1), 500.

Magidson JF, Joska J, Regenauer KS, Satinsky E, Andersen L, Borba CPC, et al. "Someone who is in this thing that I am suffering from": The role of peers and other facilitators for task sharing substance use treatment in South African HIV care. International Journal of Drug Policy. 2018.

Magidson JF, Andersen L, Satinsky EN, Myers B, Kagee A, Joska JA. "Too much boredom isn't a good thing": Adapting behavioral activation for substance use in a resource-limited South African HIV are setting. Psychotherapy. In press.

Myers, B., Carney, T., Browne, F. A., & Wechsberg, W. M. (2019). A trauma-informed substance use and sexual risk reduction intervention for young South African women: a mixed-methods feasibility study. BMJ open, 9(2), bmjopen-2018.

Myers, B., Petersen-Williams, P., van der Westhuizen, C., Lund, C., Lombard, C., Joska, J. A., ... & Stein, D. J. (2019). Community health worker-delivered counselling for common mental disorders among chronic disease patients in South Africa: a feasibility study. BMJ open, 9(1), e024277.

Myers, B., Sorsdahl, K., Morojele, N. K., Kekwaletswe, C., Shuper, P. A., & Parry, C. D. (2017). “In this thing I have everything I need”: perceived acceptability of a brief alcohol-focused intervention for people living with HIV. AIDS care, 29(2), 209-213.

Sorsdahl, K., Myers, B., Ward, C. L., Matzopoulos, R., Mtukushe, B., Nicol, A., ... & Stein, D. J. (2015). Adapting a blended motivational interviewing and problem-solving intervention to address risky substance use amongst South Africans. Psychotherapy research, 25(4), 435-444.

4. The authors should describe in detail and provide in the appendices their pleasant events schedule since the original one cited in the manuscript is very inappropriate for this population.

RESPONSE: We agree that ensuring the relevance and appropriateness of the pleasant events schedule for this population is an important consideration, and that the original version of the Pleasant Events Schedule developed in the treatment for depression in a higher income sample has items that are inappropriate for this context. 

We now include our adapted Pleasant Events Schedule in Appendix A. 

p. 9: “We used an adapted version of the Pleasant Events Schedule that we had previously modified for an underserved, substance using population in a low-income setting and further adapted for this setting.” 

5. Provide information on the amount of reimbursement also provide more details on the focus group participants for the peer recovery coaches.

RESPONSE: We have now provided more information on reimbursement and more details on focus group participants for the peer recovery coaches as suggested (Table 1).

We now specify that reimbursement was in the form of a gift card, with KI participants receiving $10 and FG participants receiving $50 gift cards. Reimbursement was determined based upon expected time commitment and was designed to be consistent with other research conducted at this community site.

p. 9: “Clients were offered gift certificates as reimbursement for their time ($10). Reimbursement was determined based upon expected time commitment and for consistency with other research conducted at this community site.” 

p. 10: “FG participants were offered gift certificates as reimbursement for their time ($50). Reimbursement was determined based upon expected time commitment and for consistency with other research conducted with staff and PRCs at this and other local community sites.” 

To provide more detail on the focus group participants for the peer recovery coaches, under “Focus group procedures” (p. 9-10) we now include a sentence outlining recruitment procedures for community PRCs: “The PRC hired for the study helped recruit community PRCs from local agencies, health care organizations, and local peer recovery networks.” As described under “Participants” (p. 12), PRCs “had experience in a range of substance use treatment environments including the emergency department of a local hospital, a prison, a community resource center, and a mental health and substance use treatment center. 

We have also included a table outlining demographic characteristics of client, staff, and community PRC participants (Table 1, p. 12), and we have separated out the characteristics of the PRCs in response to this comment. 

6. Given the huge differences in environments between impoverished and more affluent communities, if the focus of the study was Behavioral Activation, what were the recommendations and findings on how such an approach can be adapted to urban inner-cities with high levels of crime, open-air drug markets, drug use within families, low levels of employment, and few economic resources. Some of this material appears in the quotes, but it is a critical factor for adapting Behavioral Activation to this population using peer recovery coaches.

RESPONSE: We agree this is a very important consideration, and one we have considered in our team’s work adapting behavioral activation (BA) for underserved communities experiencing poverty globally and locally (see references below). We have now cited this work and discussed key considerations in adapting BA in urban inner-cities with high levels of crime, open-air drug markets, drug use within families, low levels of employment, and few economic resources. 

During the interviews and focus group, this environmental instability, including open-air drug markets and crime, repeatedly arose as a barrier to recovery and a successful peer recovery coach (PRC)-delivered BA intervention (p. 17-19). For example, a participant on p. 18 described the constant reminders of drugs and drug paraphernalia: there’s a “needle here and a needle there”. Other participants reiterated concerns around high crime neighborhoods presenting further barriers to recovery and engagement in the proposed intervention model. 

We recognize the importance of acknowledging these community-level factors and resource constraints in identifying and scheduling activities, with a focus on unique risk factors for relapse, safety when engaging in activities, and identifying individualized activity scheduling plans based on these unique risk factors. We have added this detail on p. 25: 

“When adapting a BA intervention specifically, it is important to acknowledge community-level factors and resource constraints in identifying and scheduling activities, with a focus on unique risk factors for relapse, safety when engaging in activities, and identifying individualized activity scheduling plans that take these risk factors into account.” 

Additional references were added on adapting BA for substance use in low-income, urban settings: 

Daughters SB, Magidson JF, Anand D, Seitz-Brown CJ, Chen Y, Baker S. The effect of a behavioral activation treatment for substance use on post-treatment abstinence: a randomized controlled trial. Addiction. 2017;113(3):535-44.

Magidson JF, Gorka SM, MacPherson L, Hopko DR, Blanco C, Lejuez CW, et al. Examining the effect of the Life Enhancement Treatment for Substance use (LETS ACT) on residential substance abuse treatment retention. Addictive Behaviors. 2011;36:615-23.

Magidson JF, Seitz-Brown CJ, Safren SA, Daughters SB. Implementing behavioral activation and life-steps for depression and HIV medication adherence in a community health center. Cognitive and Behavioral Practice. 2014;21:386-403.

Daughters SB, Magidson JF, Schuster RM, Safren SA. ACT HEALTHY: A combined cognitive-behavioral depression and medication adherence treatment for HIV-infected substance users. Cognitive and Behavioral Practice. 2010.

7. One of the themes was “staying busy” but this seem different, albeit some overlap, with pleasant events.

RESPONSE: We appreciate this comment, and now describe overlap and differences between “staying busy” and BA in the Discussion. We note that “staying busy” emerged as a theme in the interviews and that, while some research suggests that staying busy is linked to positive substance use outcomes (references included below and now added on p. 23), it is important to make the distinction that BA focuses on identifying activities that are rewarding and values-driven for each individual. We have provided this clarification on p. 23: 

“Some client participants described the role of “staying busy” in curbing cravings and preventing relapse. While there is some research to suggest that daily routinization and scheduling (or staying busy) can be beneficial for substance use outcomes, the conceptual model for BA encourages identification of activities that are uniquely rewarding and values-driven for the individual.”

Magidson JF, Blashill AJ, Safren SA, Wagner GJ. Depressive symptoms, lifestyle structure, and ART adherence among HIV-infected individuals: A longitudinal mediation analysis. AIDS and Behavior. 2015;19(1):34-40.

Wagner GJ, Ryan GW. Relationship between routinization of daily behaviors and medication adherence in HIV-positive drug users. AIDS Patient Care and STDs. 2004;18(7):385-93.

8. Was “case management” a theme mentioned by clients as well as staff? The quote by a client suggests that they needed resources and support. Why would this need to be case management? How much of this could be done by peer recovery coaches without case managers? Isn’t one of the rationales for peer recovery coaches to reduce costs?

RESPONSE: This is an important point and we appreciate the opportunity to clarify. While several clients described that they needed more resources and support, they did not mention “case management” specifically. We now clarify that this additional support does not have to come from case management in the traditional sense, and rather, that PRCs could take on some of the responsibilities normally assumed by case managers (support with attaining identification and housing; connection with educational and employment services). At the same time, we recognize that for patients with more severe concerns and complex needs, ensuring support for supervision, consultation, and stepped-care models to still refer to higher levels of care is crucial for PRCs to be effective, sustainable, and promote patient safety. 

In the results section, under “PRC-delivered resource- and case-management”, we clarify that PRCs can deliver these services (which would typically fall to a case manager). In the Discussion (p. 25-26), we added: “By task sharing case management services with PRCs and incorporating linkage to care alongside PRC-delivered BA…” 

Furthermore, we describe how our future work will evaluate whether PRCs can effectively take over case management responsibilities.

p. 26: “Subsequent phases of this work will test whether peers can indeed take over case management responsibilities while maintaining ongoing support from and linkage channels with specialized case management services for more severe concerns.” 

Reviewer #2: 

This is an interesting piece of qualitative research and is well-written. Addressing the following concerns would strengthen the article.

RESPONSE: Thank you. We appreciate the kind remarks. 

Introduction:

1. It would be valuable at some point in the introduction to provide a definition of problematic substance use and substance use disorders, and any prevalence data (in addition to numbers of opioid-related deaths)

RESPONSE: Thank you for this important point, and we agree this is valuable information to include. We have now included data on the prevalence of substance use disorder in Maryland and among homeless residents of Baltimore City in the Introduction on p. 3:

“While National Survey on Drug Use and Health data indicate that 7.9% of Maryland residents met criteria for SUD in 2017, an estimated 44% of people experiencing homelessness in Baltimore City are living with a SUD.” 

We have also now more clearly defined and distinguished between problematic substance use and substance use disorders in the Methods on p. 7:

“For the purposes of this study, we based inclusion criteria on problematic substance use as opposed to a diagnosis of SUD to reflect the priorities in this setting and to increase the reach of the intervention. Problematic substance use was defined based on the “moderate” to “severe” category of the WHO-ASSIST (45), which is a well-validated screening assessment used to identify problematic substance use and determine severity. Moderate to severe problematic substance use on the WHO-ASSIST reflects use that may present an individual with health, social, or economic problems, which is distinct from but may also reflect a diagnosable SUD. On the WHO-ASSIST (45), a score of 4 or above for drug use and/or a score of 11 or above for alcohol use indicates moderate to severe problematic substance use. These scores were used as cut points for study eligibility criteria.” 

2. Line 55: state some of the barriers that literature has shown that people who use substances face in accessing care.

RESPONSE: We agree that including the previously identified barriers people who use substances face in accessing care is important. We now include the following sentence in the Introduction on p. 3: “These individuals are often not engaged in the healthcare system and do not seek substance use treatment due to perceived stigma and discrimination, lack of awareness of available services, lack of healthcare coverage, and transportation and other cost barriers.” 

We include the following references: 

Carusone SC, Guta A, Robinson S, Tan DH, Cooper C, O'Leary B, et al. "Maybe if I stop the drugs, then maybe they'd care?"--hospital care experiences of people who use drugs. Harm Reduction Journal. 2019;16(16).

Liebling EJ, Yedinak JL, Green TC, Hadland SE, Clark MA, Marshall BDL. Access to substance use treatment among young adults who use prescription opioids non-medically. Substance Abuse Treatment, Prevention, and Policy. 2016;11(38).

van Boekel LC, Brouwers EP, van Weeghel J, Garretsen HF. Stigma among health professionals towards patients with substance use disorders and its consequences for healthcare delivery: systematic review. Drug and Alcohol Dependence. 2013;131(1-2):22-35.

Facing Addiction in America: The Surgeon General’s Report on Alcohol, Drugs, and Health. U.S. Department of Health & Human Services; 2016. 

3. Behavioral activation: it will be useful to provide some more information about the process and goals of behavioral activation so that it clearly links up to some of the activities in the methods section such as the completion of the readiness to change and PES exercises.

RESPONSE: We agree with this suggestion and have now elaborated on the role of BA in substance use treatment, describing the conceptual rationale for BA applied to substance use—how engagement in positive activities that are incompatible with substance use helps increase positive reinforcement and motivation for change, and disrupt negative behavioral cycles. This information has been added to the Introduction on p. 4: 

“When applied to substance use, BA encourages individuals to identify and schedule activities seen as incompatible with substance use; through these activities, individuals increase activation and positive reinforcement in their environment, develop motivation for change, and disrupt negative behavioral cycles with healthier behaviors.” 

Methods:

4. This article is an adaption of existing evidence-based interventions. Did the authors consider the use of a framework to guide this process?

RESPONSE: Thank you for the opportunity to clarify our guiding framework. This qualitative study was guided by the first three steps of the ADAPT-ITT framework. ADAPT-ITT, originally developed for HIV-related EBIs, sequentially guides the researcher through steps to adapt an intervention. The first three steps include: Assessment, Decision, and Administration. We now describe ADAPT-ITT and our use of these first three steps throughout the Methods section. At the start of the methods we describe: 

p. 6: “This study was guided by the first three steps of the ADAPT-ITT framework: Assessment, Decision, and Administration. ADAPT-ITT, originally developed for HIV-related EBIs, sequentially lays out steps for intervention adaptation. The first step, “Assessment”, involves conducting formative research to determine stakeholders’ needs and preferences, and identify implementation barriers and facilitators. The next step, ‘Decision”, involves incorporating stakeholders’ feedback into decision-making around which intervention components to include. The third step, “Administration”, uses “theatre-testing” to pretest intervention components and elicit stakeholder reactions. This framework increases the likelihood that the intervention is acceptable and appropriate for a new setting and target population.” 

5. Since the ASSIST measures substance use in the previous three months, expand on how past and present problematic substance use were defined in this study.

RESPONSE: Thank you for this opportunity to clarify this point. We now specify that participants were asked about lifetime and past-three month use for each question on the WHO-ASSIST. This information is included on p. 7. 

6. Evaluation of ability to provide consent is important, could this be expanded upon in a sentence (what did this process look like?)

RESPONSE: We agree, and we now elaborate on the procedures for determining ability to give consent on p. 8. Specifically, we describe: 

“After the research team member read the consent form aloud, the client was asked to respond to a series of questions meant to guide the evaluation of one’s ability to provide consent. Specifically, the participant was asked to reiterate (1) what would be expected of him or her, (2) potential risks, and (3) what they would do if they decided they no longer wished to participate. If a client struggled to answer one of these questions, the research team member reviewed the study procedures until the client was able to explain expectations. If the client was unable to adequately answer these questions, even after reviewing the consent form with the research team member, they were determined unable to provide consent.”

7. There a few acronyms that are only used once or twice in the article such as DBT, LTC and OUD. It would be easier to read if these were spelled out in the manuscript.

RESPONSE: We agree, and we have now removed the acronyms DBT, LTC, and PES. We instead spell out what they stand for. Since ‘OUD’ is used more frequently in the text, we have left this acronym in; however, we have clearly spelled out OUD as opioid use disorder the first time it is used in the text. 

8. Explain why the focus group took part after half of the interviews had been completed

RESPONSE: Thank you for the opportunity to clarify this decision. We planned to hold the focus group after at least half of the interviews had been completed to be able to provide a summary of main themes from the client interviews up until that point to get staff feedback and responses. We also wanted to then be able to incorporate staff focus group feedback into remaining individual interviews with clients, including the opportunity to adapt the interview guides if necessary. This is described on p. 10:

“The focus group took place after the first half of client interviews had been conducted; as such, the research team had an initial sense of client responses to the proposed intervention and were able to adapt the focus group questions and structure appropriately.” 

9. What were the value of the gift certificates provided for reimbursement.

RESPONSE: We now specify that KI participants received $10 and FG participants received $50 gift certificates for their time. Reimbursement was determined based upon expected time commitment and consistency with other research conducted at this community site (p. 9-10). 

10. Please clarify when participants were screened for eligibility and why one did not meet eligibility criteria AFTER consenting took place.

RESPONSE: As we now clarify earlier in the Methods on p. 7-8, consent took place prior to the screening procedures. We further note at the top of p. 7 that “Research staff discussed the study procedures with center clients, briefly describing aims and explaining the focus on individuals with experiences of substance use.” As such, clients who expressed interest in participating in an interview were aware of the focus on substance use.

Three participants who were consented did not end up participating in the interviews:

p. 10: “…one decided she was no longer interested after completing the consent process, and two did not meet eligibility criteria (n=1 did not meet criteria for moderate to severe past or present problematic substance use on the WHO-ASSIST; n=1 showed signs of active psychosis during screening procedures that would interfere with participation in the semi-structured interview).” 

11. Think about possibly providing a table to summarize the KI and FG participant characteristics. Also, since opioid use is clearly an issue in this setting, can the authors provide the proportion of problem opioid use in general (in addition to the average score on the ASSIST?) 

RESPONSE: We now provide a Table (Table 1 on p. 12) to describe the KI and FG participant characteristics. We also report the percent of all participants with past or present problematic cocaine use (77%, n=23) and past or present problematic opioid use (70%, n=21). We provide a breakdown of the substance use risk categories for alcohol, cannabis, cocaine, and opioids (stratified by past and present use) in Table 2 (p. 12). 

Results:

12. Can the authors indicate when the findings and quotes were from the FGs and when they were from the KI interviews in the results section?

RESPONSE: Yes, now each time we reference a new participant quote, we include whether they were from the key informant interviews (KI) or focus group (FG). 

13. Some of the quotes back up the same argument and can be put together without further text in between. For example, line 254 on page 12 and lines 282-283 on page 13 can be removed.

RESPONSE Thank you. These lines have been removed. 

14. Line 265-267 on page 12: rather use such as than abbreviations i.e. and etc.

RESPONSE: We have replaced the abbreviations ‘i.e.’ and ‘etc.’ with ‘such as’ or ‘including’. 

15. The voice of the PRCs seem to be somewhat missing in the quotes section, are there any quotes are any interesting points that they raised specifically as opposed to other staff in the centers?

RESPONSE: We are grateful that the reviewer made this suggestion and appreciate the opportunity to provide more feedback directly from the PRCs, as they did indeed have a unique perspective from other staff and raised interesting points. We have now added quotes from PRCs. For instance, when discussing participants’ preference for working with a PRC, we include the following quote from a PRC FG participant on p. 14: 

“’I’ll start to disclose or share part of my story then. And it kinda, it pretty much draws them in. Cause now, it’s, there’s no hierarchy. We’re on a level plane.’” – [PRC FG Participant (P2)]

Additionally, on p. 21 in the section, “PRC-delivered resource management”, we include the following quote from a PRC, illustrating how they can help address barriers to care: 

“Keep them linked up…to their treatment and whatever other type of assistance I can help them with. Like ID, maybe a job lead, maybe housing, food, etc.” – [PRC FG Participant (P3)]

16. P17 line 364-I am not sure that this quote is necessary.

RESPONSE: Although we have left this quote in the manuscript, we hope that the addition of the above-mentioned quote (“Keep them linked up…”; p. 21) helps to further contextualize this statement by a staff FG participant. 

17. P17 line 368-not entirely sure what this quote is saying-are there any other quotes where the types of resources are discussed that the authors could substitute here?

We have removed this quote and hope that the above-mentioned PRC quote provides information on the types of resources discussed (p. 21): “Keep them linked up…to their treatment and whatever other type of assistance I can help with. Like ID, maybe a job lead, maybe housing, food, etc.” – [PRC FG Participant (P3)]

Discussion:

18. P18 line 399: "Yet, clients and staff/PRCs presented barriers to consider." This sentence feels incomplete.

RESPONSE: Thank you. This sentence has been revised to read: “Despite positive responses, clients and staff/PRCs presented barriers to consider when adapting a PRC-delivered intervention.” 

19. P19: Are there other studies conducted with PRCs outside of Baltimore than can be references here?

RESPONSE: We have now added on p. 23 that studies conducted in other settings in the United States and globally have similarly reflected the acceptability and positive effects of peer-delivered services for substance use: 

“Studies conducted in emergency departments and clinical settings in the US and in other contexts globally have similarly reflected the acceptability and potential for peers in harm reduction and recovery support services, including for reduction of use, decreased risk behaviors, and improved quality of life.” 

We include the following references, which reflect research from across the United States, China, Vietnam, and Malaysia: 

Jack HE, Oller D, Kelly J, Magidson JF, Wakeman SE. Addressing substance use disorder in primary care: The role, integration, and impact of recovery coaches. Substance Abuse. 2017:1-8.

Eddie D, Hoffman L, Vilsaint C, Abry A, Bergman B, Hoeppner B, et al. Lived experience in new models of care for substance use disorder: A systematic review of peer recovery support services and recovery coaching. Frontiers in psychology. 2019;10(1052).

Hammett TM, Jarlais DCD, Kling R, Kieu BT, McNicholl JM, Wasinrapee P, et al. Controlling HIV Epidemics among Injection Drug Users: Eight Years of Cross-Border HIV Prevention Interventions in Vietnam and China. Plos One. 2012;7(8).

Rashid RA, Kamali K, Habil MH, Shaharom MH, Seghatoleslam T, Looyeh MY. A mosque-based methadone maintenance treatment strategy: implementation and pilot results. The International journal on drug policy. 2014;25(6):1071-5.

Samuels EA, Baird J, Yang ES, Mello MJ. Adoption and utilization of an emergency department naloxone distribution and peer recovery coach consultation program. Academic Emergency Medicine. 2019;26(2):160-73.

Samuels EA, Bernstein SL, Marshall BDL, Krieger M, Baird J, Mello MJ. Peer navigation and take-home naloxone for opioid overdose emergency department patients: Preliminary patient outcomes. Journal of Substance Abuse Treatment. 2018;94:29-34.

Scott CK, Dennis ML, Grella CE, Kurz R, Sumpter J, Nicholson L, et al. A community outreach intervention to link individuals with opioid use disorders to medication-assisted treatment. Journal of Substance Abuse Treatment. 2019.

Scott CK, Grella CE, Nicholson L, Dennis ML. Opioid recovery initiation: Pilot test of a peer outreach and modified Recovery Management Checkup intervention for out-of-treatment opioid users. Journal of Substance Abuse Treatment. 2018;86:30-5.

Watson DP, Brucker K, McGuire A, Snow-Hill NL, Xu H, Cohen A, et al. Replication of an emergency department-based recovery coaching intervention and pilot testing of pragmatic trial protocols within the context of Indiana's Opioid State Targeted Response plan. Journal of Substance Abuse Treatment. 2019.

20. P19 line 420: There are a number of studies that look at physical activity and religion as prosocial activities that are alternatives to substance use including studies conducted in the US and globally. Could the authors do some additional research and reference some of these?

RESPONSE: Thank you for this suggestion. We have brought in a broader literature to support physical activity and religion as prosocial activities that may be alternatives to substance use. 

p. 24: “These specific categories are consistent with research describing the role of religious activity, including associated spirituality and social support, and sports and exercise, in supporting recovery from problematic substance use. Furthermore, these reflect the potentially reinforcing, substance-free activities listed in qualitative interviews among a population with problematic substance use in a South African HIV care setting, indicating the potential role of these activities in BA for problematic substance use across cultures.” 

The following references are now included and reflect research from Brazil, South Africa, the United States, and beyond.

Petry NM, Lewis MW, Ostvik-White EM. Participation in religious activities during contingency management interventions is associated with substance use treatment outcomes. American Journal on Addictions. 2009;17(5):408-13.

van der Meer Sanchez Z, Nappo SA. Religious intervention and recovery from drug addiction. Revista de Saude Publica. 2008;42(2).

Laudet AB, Morgen K, White WL. The role of social supports, spirituality, religiousness, life meaning and affiliation with 12-Step fellowships in quality of life satisfaction among individuals in recovery from alcohol and drug problems. Alcoholism Treatment Quarterly. 2006;24(1-2):33-73.

Francis JM, Myers B, Nkosi S, Petersen Williams P, Carney T, Lombard C, et al. The prevalence of religiosity and association between religiosity and alcohol use, other drug use, and risky sexual behaviours among grade 8-10 learners in Western Cape, South Africa. PLoS One. 2019;14(2):e0211322.

Wang D, Wang Y, Wang Y, Li R, Zhou C. Impact of physical exercise on substance use disorders: A meta-analysis. PLoS One. 2014;9(10):e110728.

Zschucke E, Heinz A, Strohle A. Exercise and physical activity in the therapy of substance use disorders. The Scientific World Journal. 2012;2012.

21. It is interesting that the majority of the participants are homeless-how will this affect their access to engagement in interventions, as this is different to living in economically disadvantaged areas with high levels of crime.

We agree this is a very important consideration, and one we must consider based on the high rate of homelessness seen among this population. We now note the additional barrier some individuals may face in accessing treatment and participating in BA due to homelessness and housing instability. We note that peer-delivered linkage to treatment may include connection with intensive outpatient treatment programs that include a residential housing component and support attainment of housing on p. 25-26.

“Furthermore, considering the high rate of homelessness and housing instability seen among this population, individuals may face additional barriers to engaging in care and incorporating positive reinforcing, values-driven activities into their daily lives…Among individuals concurrently experiencing homelessness and severe problematic substance use, linkage may involve connection with intensive outpatient treatment programs that include a residential housing component and support attainment of permanent housing following treatment completion. Subsequent phases of this work will test whether peers can indeed take over case management responsibilities while maintaining ongoing support from and linkage channels with specialized case management services for more severe concerns.” 

22. Since there are very likely participants that had levels of substance use that placed them at high levels of risk, it is possible that they may need more formalized psychosocial treatment in addition to pharmacological assistance. This is something for the authors to consider in the discussion.

RESPONSE: We absolutely agree with this comment and we appreciate the opportunity to more clearly highlight this point in the Discussion. We have now expanded on the importance of linkage to MOUD as well as formalized psychosocial treatment programs for co-occurring substance use and mental health problems. 

On p. 26 we elaborate: “Finally, importantly for patients with OUD, PRC-delivered linkage to care could include referring clients to MOUD, as well as other formalized psychosocial treatment programs for co-occurring substance use and mental health problems.” 

We also describe on p. 27 that “In the ongoing research guided by these qualitative results, we aim to explore how PRCs can support delivery of psychosocial treatment, under the close supervision and monitoring of licensed mental health professionals, to support improved MOUD outcomes.”

---

## [Decision Letter · Decision Letter 1]

8 Jan 2020

Adapting a community-based peer recovery coach-delivered behavioral activation intervention for problematic substance use in a medically underserved community in Baltimore City

PONE-D-19-16724R1

Dear Dr. Satinsky,

We are pleased to inform you that your manuscript has been judged scientifically suitable for publication and will be formally accepted for publication once it complies with all outstanding technical requirements.

With kind regards,

Bronwyn Myers

Academic Editor

PLOS ONE

Additional Editor Comments (optional):

Thank you for submitting this revised manuscript- I am pleased to inform you that the reviewers are happy with the revisions and your paper is now accepted for publication. Congratulations.

Reviewers' comments:

Reviewer's Responses to Questions

**Comments to the Author**

1. If the authors have adequately addressed your comments raised in a previous round of review and you feel that this manuscript is now acceptable for publication, you may indicate that here to bypass the “Comments to the Author” section, enter your conflict of interest statement in the “Confidential to Editor” section, and submit your "Accept" recommendation.

Reviewer #1: All comments have been addressed

2. Is the manuscript technically sound, and do the data support the conclusions?

Reviewer #1: Yes

3. Has the statistical analysis been performed appropriately and rigorously? 

Reviewer #1: Yes

4. Have the authors made all data underlying the findings in their manuscript fully available?

Reviewer #1: Yes

5. Is the manuscript presented in an intelligible fashion and written in standard English?

Reviewer #1: Yes

6. Review Comments to the Author

Reviewer #1: (No Response)

7. PLOS authors have the option to publish the peer review history of their article (what does this mean?). If published, this will include your full peer review and any attached files.

Reviewer #1: No

---

## [Editor Report · Acceptance letter]

10 Jan 2020

PONE-D-19-16724R1 

Adapting a peer recovery coach-delivered behavioral activation intervention for problematic substance use in a medically underserved community in Baltimore City 

Dear Dr. Satinsky:

I am pleased to inform you that your manuscript has been deemed suitable for publication in PLOS ONE. Congratulations! Your manuscript is now with our production department. 

With kind regards,

on behalf of

Dr. Bronwyn Myers 

Academic Editor

PLOS ONE